# A New Species in *Notobitiella* Hsiao in China Confirmed by Morphological and Molecular Data (Hemiptera: Heteroptera: Coreidae)

**DOI:** 10.3390/insects13050411

**Published:** 2022-04-26

**Authors:** Kun Jiang, Juhong Chen, Wenjun Bu

**Affiliations:** College of Life Sciences, Nankai University, Tianjin 300071, China; jiangkunnk@163.com (K.J.); 15181674153@163.com (J.C.)

**Keywords:** Cloresmini, *Notobitiella*, monotypic genus, species delimitation by COI, bamboo

## Abstract

**Simple Summary:**

Plenty of new species are published every year, but what if a new species could be erected as a new genus by the morphological characteristics of genitalia or a new species of an existing genus by the other morphological characteristics? Recently, we collected a new species of Coreidae facing this problem. The characteristics of genitalia between the new and existing species in *Notobitiella* (monotypic genus) were too different to compare with the difference among interspecies in tribe Cloresmini. It was reasonable to erect a new genus for this new species. However, considering the similarity between other characteristics, we decided to refer to the results of molecular data. Results showed that the genetic distance within the interspecific interval of the genus and the sister group relationship also supported that the new species should be included in the genus *Notobitiella*. Finally, a new species was confirmed, named *Notobitiella bispina*
**sp. nov**. Our research highlighted how molecular data can be used to assist in species delimitation when encountering similar problems.

**Abstract:**

To date, only one species of genus *Notobitiella*, *N. elegans* Hsiao, was found in Yunnan, China. Recently, we confirmed a new species, *Notobitiella bispina*
**sp. nov**., by morphological and molecular data based on new specimens collected from Yunnan, China. The new species is morphologically similar to *N. elegans* except for the male genitalia and the sternum of the seventh abdominal segment of the female. The extraordinary difference of the genitalia between these two species inspired us to erect a new genus for this new species. However, considering their sister group relationship to other genera in tribe Cloresmini and the 12.56~12.64% genetic distance (meeting the interspecific genetic distance within genus of tribe Cloresmini) by a complete COI gene, this species was more reasonable as a new species of the genus *Notobitiella*, and the diagnosis of the genus was revised. The photos of the body and critical morphological characteristics for both male and female were provided for accurate identification.

## 1. Introduction

The genus *Notobitiella* was established by Hsiao based on one species, *Notobitiella elegans* Hsiao, 1963 [1]. At present, only one species has been recorded in this genus. This species was found in Xiaomengyang, Xishuangbanna, Yunnan, China, inhabiting in gaps of top new leaves of bamboo shoots [2,3].

A mitochondrial gene, cytochrome *c* oxidase I (COI), has been identified as an available sequence in species delimitation [4,5,6]. Park et al., 2011 found that for the researched 344 heteropteran species which applied DNA barcoding, the minimum interspecific distances exceeded 3% in 77% of congeneric species pairs [7]. Another study for 457 true bugs in Germany indicated that DNA barcoding had a successful identification rate of 91.5% [8]. Although these studies could not distinguish every species, partial COI (DNA barcoding) represents a useful marker in species delimitation [7,8,9]. Further, complete COI offered more phylogeny information which has successfully been applied in species delimitation of North American Entomobrya and Crustacea [10,11]. Therefore, the complete COI may be even more helpful in species delimitation.

In this study, we confirmed a new species of the genus *Notobitiella*, *Notobitiella bispina*
**sp. nov**., by using morphological characteristics and the sequences of a complete COI gene based on the recent collecting trip to Xishuangbanna, Yunnan, China, the same locality with the type species of the genus.

## 2. Materials and Methods

All specimens were collected on bamboos and preserved in 98% ethanol in the field. Some were stored in −20 °C refrigerator in lab for molecular analysis, and others were dried and pinned. Photographs were taken with a Canon EOS 5D Mark II camera and synthesized with Helicon software. Measurements were taken under ZEISS SteREO Discovery V8 by vernier caliper. The type specimens were deposited in the insect collection of Institute of Entomology, College of Life Sciences, Nankai University, Tianjin, China (NKU). Morphological terminology follows Hsiao’s paper (1963b).

In addition to species of the genus *Notobitiella*, for molecular analysis, we also collected species of the genus *Notobitus* and *Cloresmus* in the tribe Cloresmini of China and Laos. Detailed information about the specimens is presented in Table 1. As all samples were collected from the same location as the species already presented, females and males of each species were selected for sequencing to ensure a correct match between females and males. Genomic DNA was extracted from thorax muscles using a DNeasy Blood & Tissue Kit (QIAGEN). For each sample, the whole mitochondrial genome was sequenced using the Illumina HiSeq 2000 platform (Illumina) with a 350 bp insert size and a paired-end 150 bp sequencing strategy at Novogene. A total of 2 GB raw data were obtained for each sample. Clean data were assembled by MitoZ and annotated by MITOS web server [12] (http://mitos.bioinf.uni-leipzig.de/index.py/, accessed on 20 November 2021). Finally, the annotated mitochondrial genome was carefully checked and revised by ORFfinder of NCBI (https://www.ncbi.nlm.nih.gov/orffinder/, accessed on 25 November 2021). The complete COI gene for all samples were extracted and alignment of the sequences was carried out using the MUSCLE algorithm [13] on amino acids in MEGA X [14]. The complete sequence of the COI gene for *Cloresmus pulchellus* and three outgroups *Clavigralla tomentosicollis*, *Leptoglossus membranaceus, and Hydaropsis longirostris* from Liu et al., 2019 [15,16,17] were also included in our analysis (GenBank Accession Number: NC_042806, KY274846, NC_042809, and EU427337). The genetic distance matrix estimations and neighbor-joining tree [18] based on COI datasets were calculated in MEGA X, using the Kimura 2-Parameter (K2P) substitution model [19], 1000 non-parametric bootstrap replicates [20], and the “pairwise deletion” option for missing data [21].

## 3. Results

Genera in Cloresmini could be distinguished by the characteristics of fore femur, body shape and color, the length of rostrum, and other characteristics. The new species could be easily grouped to monotypic genus *Notobitiella* by the aforementioned characteristics. Nevertheless, the genitalia of the new species were different than those of *N. elegans*’ (Figure 1 and Figure 2). In order to evaluate if a new species or a new genus was found, complete COI for species delimitation was analyzed.

Neighbor-joining tree based on the complete COI gene of six morphospecies within the tribe Cloresmini revealed that the new species and *N. elegans* formed a monophyletic group, representing the sister group to the genera *Cloresmus* and *Notobitus* (Figure 3). The new species had a maximum intraspecific pairwise genetic distance of 0.13% and a minimum interspecific genetic distance of 12.56% to *N. elegans* (Table 2). Interspecific genetic distance within genera in tribe Cloresmini was 12.30~13.55% and the intergeneric genetic distance in tribe Cloresmini was 12.12~15.82%. In addition, the COI gene for different genders showed that the match of females and males for both species was correct.


**Taxonomy**



**Genus *Notobitiella* Hsiao, 1963**


*Notobitiella* Hsiao,1963b:322–323; Hsiao et al., 1977:224.

**Type species.***Notobitiella elegans* Hsiao, 1963


**Revised diagnosis.**


When Hsiao (1963) established the genus *Notobitiella*, he found that the main differences between *Notobitiella* and other two genera, *Notobitus* and *Cloresmus*, in tribe Cloresmini were the lengths of antennae and rostrum, the spines on the hind femur, and the characteristics of male genitalia and structure of the sternum of the 7th abdominal segment of the female. After we added this new species, we revised the generic diagnosis as follows:

Body elongated, cylindrical in shape, with dull metallic iridescence on the dorsal side; rostrum reached the base of middle coxae, basal segment did not reach the trailing edge of the eyes; the dorsal side of hind femur was blackish-blue with metallic iridescence, covered with rows of short furcellas, ventral side dark brown.


**Description of new species.**



***Notobitiella bispina* Jiang, Chen et Bu, sp. nov.**


(Figure 1, Figure 2, Figure 4, Figure 5 and Figure 6)

urn:lsid:zoobank.org:act:CC5F0425-E17B-4EAD-AC21-408EDA98AE2E

**Type material.** Holotype: male, Mengyang, Xishuangbanna, Yunnan, China, 22.1098° N, 100.3636° E, 1071 m a.s.l., 22.XII.2021, collected from bamboo by Kun Jiang & Juhong Chen. Paratype: one male and five females pinned, two males and six females kept in 98% alcohol. Same field collection data as holotype.

**Etymology.** The specific name, *bispina*, refers to the male with two long spines on the lateral-backward side of the 7th abdominal segment.

**Description. Body type and color:** Body elongated, cylindrical in shape, black with dull blue metallic iridescence, clothed with short hairs and punctates (Figure 4). Front-end of antenniferous tubercles brown, first three segments of antennae dark red and end of second and third segments black, fourth segment of antennae dark grey. Dorsal head (or just posterior), pronotum, scutellum, thorax lateral and beneath, and coxae of all legs black with dull blue metallic iridescence. Ventral head (some samples included anterior dorsal head), rostrum, buccula, fore and middle legs and hind tarsi yellowish brown. Ocelli posterolateral head with yellowish-brown stripes. Collar dark brown to black. Surrounding areas of scent gland ostioles yellowish brown, evaporatorium dull grey. Forewings metallic luster, coria dull reddish brown, membranes metallic green. Dorsal hind femora blackish-blue with bright metallic iridescence, ventral hind femora dark red. Hind tibiae red at base, gradually lightening in color to the tip. Dorsal and ventral side of abdomen red, dorsal side of abdomen with 7th and posterior edge of fourth to sixth segments black, central joint of fourth to fifth and fifth to sixth with a light-yellow point. Connexivum light yellow, black at base. Two long spines on the lateral-backward side of the seventh abdominal segment black (only for male). Genitalia for both male and female yellowish brown. **Head:** head lateral lobes shorter than central lobe; antenniferous tubercles slightly prominent; first three segments of antennae covered with short fine hairs, first to second segments roughly equal, third segment shortest, fourth segment longest with quite short hairs; ocelli as far from one another as from eyes; buccula prominent; rostrum with short hairs, reaching apex of middle coxae, basal one not reaching base of head. **Thorax:** thorax finely punctate with short hairs; anterior lope of pronotum smooth, collar very slender, lateral margins rounded, moderately sinuate, lateral angles rounded; scutellum roughly equilateral triangle, vertex angle sharp; both middle and hind coxae remote from one another; outboard of each hind coxae with a little blunt tooth; femora incrassate, ventral side of each front and middle ones with two rows of small spines; upper side of hind femora with rows of short furcellas, ventral side with a row of relatively big spines, the biggest spine situating from the 1/3 of the way from the end; hind tibiae basal inward bend, end half outward bend, ventral side with a row of teeth, dorsal side with groove. **Abdomen:** abdomen shorter than hind femora; sides of the abdomen subparallel, edge with finely double serrate, lateral-backward of each segment with two small spines; male lateral-backward side of the seventh abdominal segment with two long black spines, point to backward, female without this trait (Figure 1A,B). **Genitalia:** male genital capsule inner with long hairs (Figure 1A and Figure 2A), posterior central edge with three little teeth for the view of about 45 degree angle of the ventral surface. In most cases, bigger in the middle, smaller on each side, but have variations for some samples (Figure 5); parameres quite small and short. Female posterior seventh abdominal segment with a “T” shape crack, without any spines. Whole genitalia with long light-yellow hairs (Figure 1B).

**Measurement.** Head length, 1.76 mm; width across eyes, 2.11 mm; interocular distance, 1.02 mm; interocellar distance, 0.42 mm; length of antennal segments: I, 3.07: mm; II, 3.65 mm; III, 3.06 mm; IV, 4.51 mm; length of rostral segments: I, 1.02 mm; II, 1.22 mm; III, 1.29 mm; IV, 1.24 mm. Pronotum. Length, 3.29 mm; maximal width of anterior lobe, 2.2 mm; maximal width of posterior lobe, 3.82 mm. Male (n = 4). Total length 15.12–17.72, 16.05 mm. Total width 3.71–4.21, 3.93 mm. Female (n = 11). 15.61–18.19, 16.57 mm. Male (n = 4). Body length 15.12–17.72, 16.05 mm, width 3.71–4.21, 3.93 mm. Female (n = 11). Body length 15.61–18.19, 16.57 mm, width 3.64–4.51, 4.03 mm.

**Habitat and biology.** We only found this species in Mengyang, Xishuangbanna, Yunnan, China. This species was found on gaps of top new leaves of bamboo shoots (Figure 6). Similar to *Cloresmus modestus* in the same tribe Cloresmini, we observed that the species in genus *Notobitiella* was also agile and flew well [22].


**Key to the species of *Notobitiella***


Head blackish-blue. Anterior hind femora dark red. Male genital capsule posterior margin broadly concave with a middle spine-like process. Parameres black, fairly long and thick, reaching out of genital capsule. Ventral side of female seventh abdominal segment with two long backward spines………………………………*Notobitiella elegans* Hsiao

Ventral and anterior head dull brown. Dorsal hind femora totally blackish-blue with dull metallic iridescence. Male seventh abdominal segment lateral-backward side with two long black spines on each side, projecting backwards. Posterior margin of male genital capsule relatively narrow and with three small teeth. Parameres light yellow, pretty small and short, didn’t reach out of genital capsule. Ventral side of female seventh abdominal segment with a “T” shape crack, without spines…………………*Notobitiella bispina*
**sp. nov.**

## 4. Conclusions

The genetic distance between the new species and *N. elegans* didn’t exceed the interspecific maximum distance within the genus, and also reached the intergeneric minimum distance in tribe Cloresmini. The new species could be erected as a new genus or a new species in genus *Notobitiella*. However, the new species and *N. elegans* represent the sister group to the genera *Cloresmus* and *Notobitus,* which indicated that this species was more reasonable to be a new species in genus *Notobitiella.* Our research did not include all genera in tribe Cloresmini, and adding the other two genera (one of them is a monotypic genus) may optimize the conclusion [23,24].

The new species of male can be separated from *Notobitiella elegans* Hsiao 1963 by having two long spines on the lateral-backward side of the seventh abdominal segment (Figure 1A,C). A notable characteristic of *N. elegans* Hsiao 1963 for the male is its short height (compared to the other species in tribe Cloresmini) and black genital capsule, posterior margin broadly concave with a middle spine-like process, and parameres that are rather long, reaching out of the genital capsule (Figure 2B). The seventh abdominal segment of the female has the posterior portion abruptly sunken, with the central fissure short and broad, produced on each side into a long process. (Figure 1D). For the new species, the genital capsule of male is narrow, with short parameres and three obscure teeth on the posterior margin of the genital capsule. (Figure 2A). The seventh abdominal segment of the female has only a crack rather than spines, such as in *N. elegans* (Figure 1B,D). In addition, the anterior and ventral side of head of *N. bispina*
**sp. nov.** is dull brown in contrast to black in *N. elegans*. The anterior hind femur of new species is black in contrast to red in *N. elegans*.

If only considering the differences of genitalia, creation of a new genus for the new species was reasonable. However, except genitalia, other morphological characteristics were fairly similar. Based on the result of molecular data and morphological characteristics, we decided to group this new species into genus *Notobitiella*. Our research highlighted that molecular data can be used to assist in species classification when encountering similar problems.

## Figures and Tables

**Figure 1 insects-13-00411-f001:**
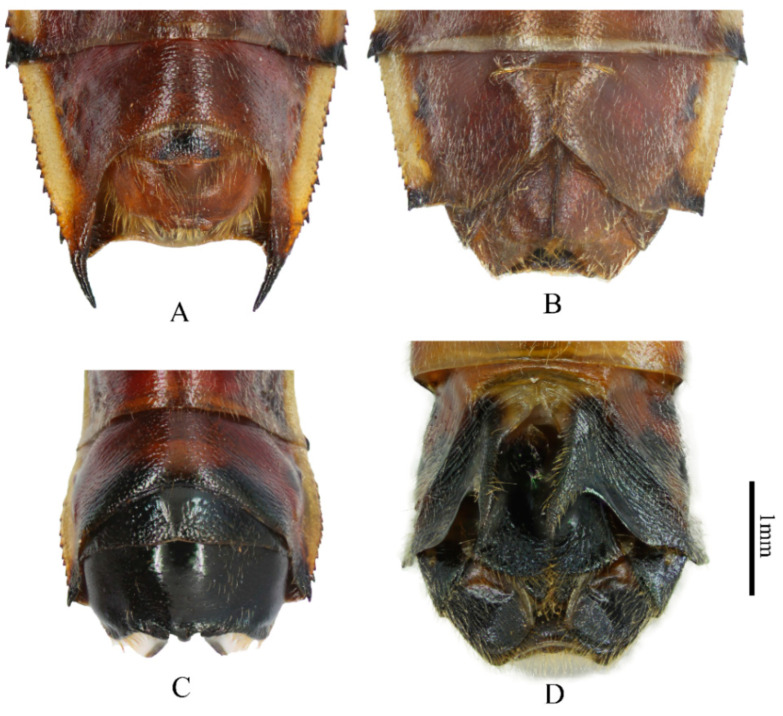
Ventral view of pygophore. (**A**) male of *N. bispina*
**sp. nov.**, (**B**) female of *N. bispina*
**sp. nov.**, (**C**) male of *N. elegans*, and (**D**) female of *N. elegans.*

**Figure 2 insects-13-00411-f002:**
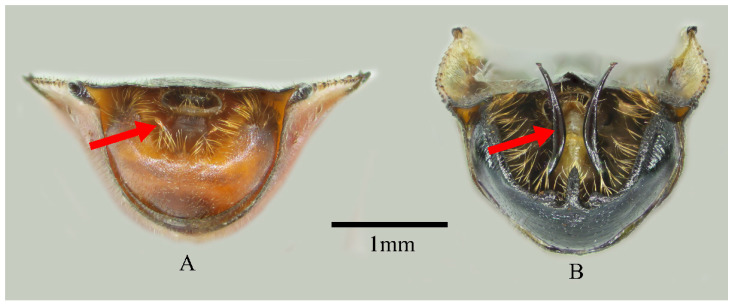
Posterior view of pygophore. (**A**) *N. bispina*
**sp. nov**., and (**B**) *N. elegans*.

**Figure 3 insects-13-00411-f003:**
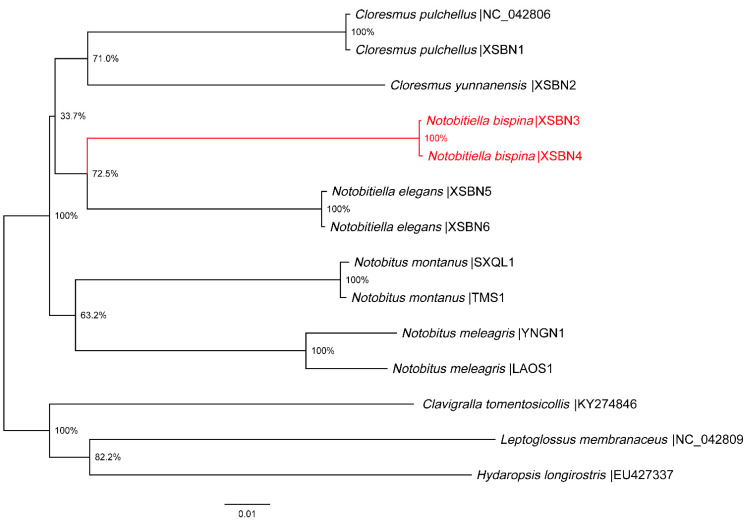
Neighbor-joining tree for six species of the tribe Cloresmini based on K2P distance in DNA barcodes. Numbers on branches represent bootstrap support based on 1000 replicates; scale equals K2P genetic distance.

**Figure 4 insects-13-00411-f004:**
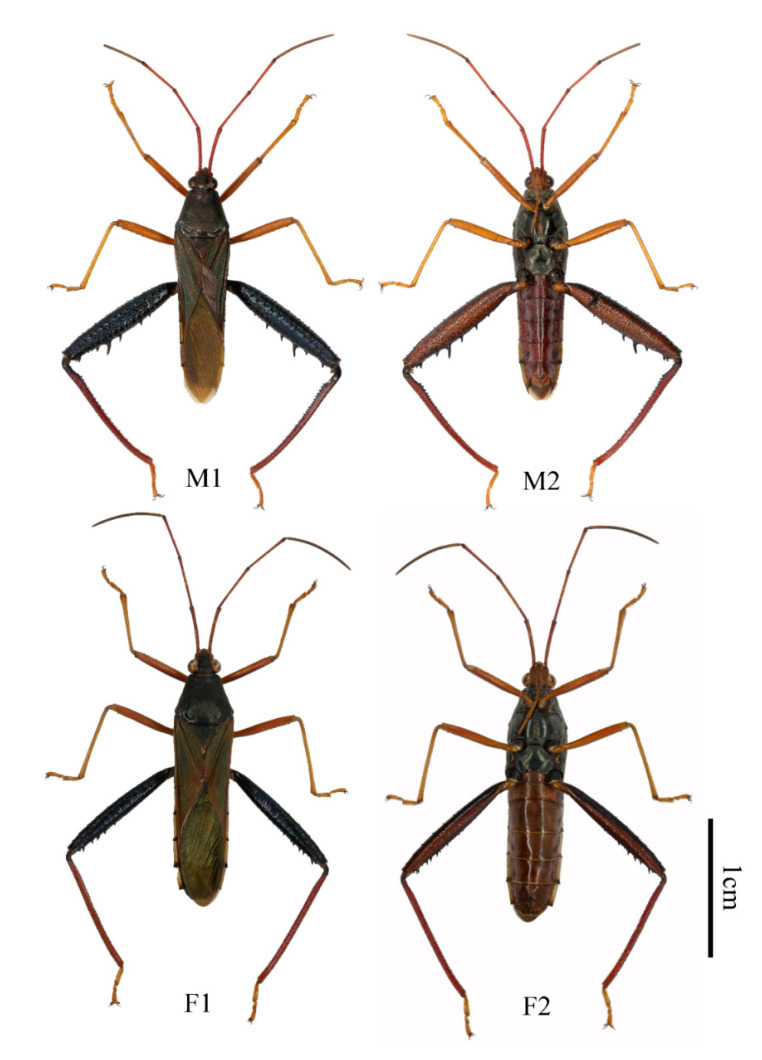
*N. bispina***sp. nov. M1** dorsal side of male **M2** ventral side of male **F1** dorsal side of female **F2** ventral side of female.

**Figure 5 insects-13-00411-f005:**
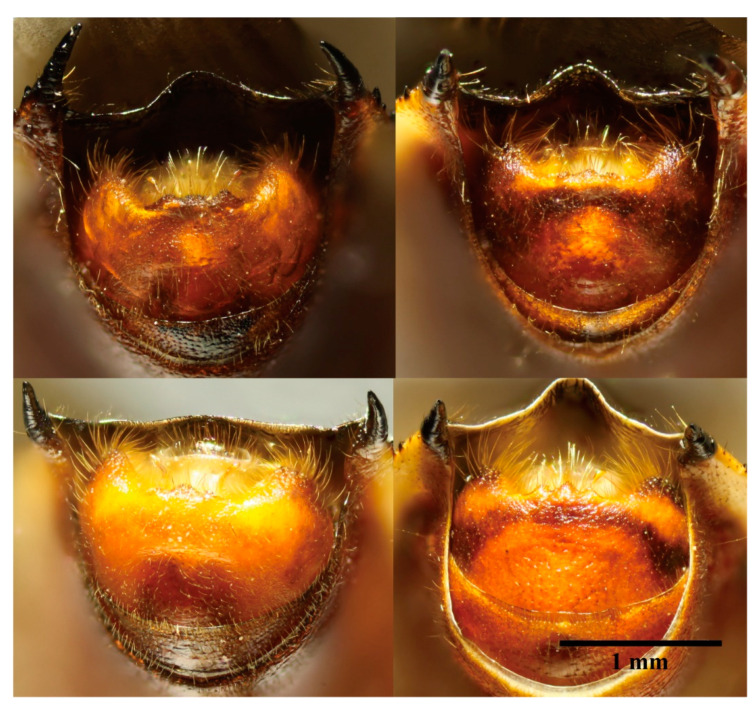
45 degree angle views of the ventral surface for four samples of the new species, *N. bispina*
**sp. nov**. showing the teeth variation on the posterior margin of male genital capsule.

**Figure 6 insects-13-00411-f006:**
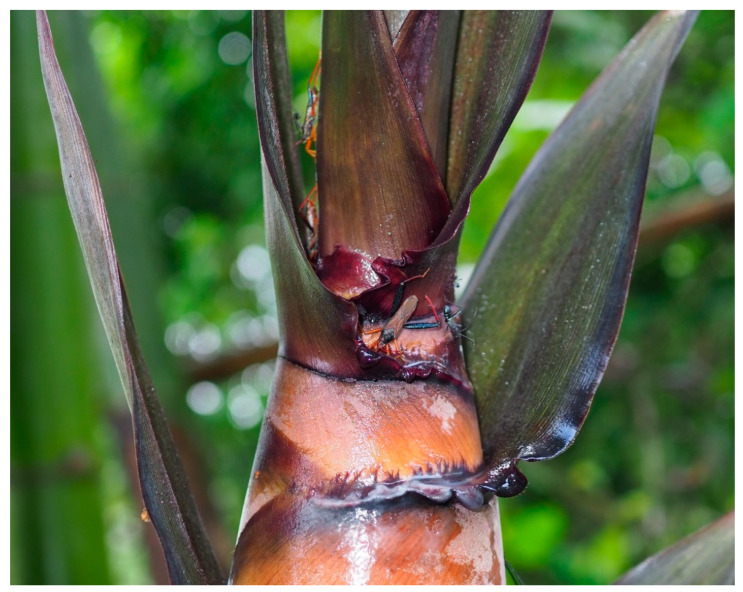
Field picture of *N. bispina*
**sp. nov.** at Xiaomengyang, Xishuangbanna, Yunnan, China.

**Table 1 insects-13-00411-t001:** Collected information of specimens used for molecular analysis.

Code	Date	Species Name	Location	Longitude	Latitude	Altitude	Collector
XSBN1	23 July 2021	*Cloresmus pulchellus*	Haiwang road, Menga, Xishuangbanna, Yunnan, China	E100.390	N22.046	1193 m	Juhong Chen
XSBN2	23 July 2021	*Cloresmus yunnanensis*	Menghun, Menghai, Xishuangbanna, Yunnan, China	E100.419	N21.869	931 m	Kun Jinag
XSBN3	23 July 2021	*Notobitiella bispina*	Haiwang road, Menga, Xishuangbanna, Yunnan, China	E100.390	N22.046	1193 m	Juhong Chen
XSBN4	23 July 2021	*Notobitiella bispina*	Haiwang road, Menga, Xishuangbanna, Yunnan, China	E100.390	N22.046	1193 m	Juhong Chen
XSBN5	22 July 2021	*Notobitiella elegans*	Haiwang road, Menga, Xishuangbanna, Yunnan, China	E100.364	N22.110	1071 m	Juhong Chen
XSBN6	23 July 2021	*Notobitiella elegans*	Haiwang road, Menga, Xishuangbanna, Yunnan, China	E100.390	N22.046	1193 m	Juhong Chen
YNGN1	9 August 2020	*Notobitus meleagris*	Babao, Guangnan, Yunnan, China	E105.495	N23.756	1046 m	Xue Dong
LAOS1	19 August 2019	*Notobitus meleagris*	Nadana Village, Salavan, Laos	E106.384	N15.717	189 m	Jiayue Zhou
SXQL1	25 June 2014	*Notobitus montanus*	Zhonghe Village, Zuoshui, Shanxi, China	E109.401	N33.569	850 m	Wenjun Bu
TMS1	26 July 2015	*Notobitus montanus*	Tianmu mountain, Zhejiang, China	E119.455	N30.329	784 m	Wenbo Yi

**Table 2 insects-13-00411-t002:** Kimura 2-parameter (K2P) pairwise genetic distances based on complete COI gene.

Species	Pairwise Genetic Distances												
*Cloresmus pulchellus*|NC_042806													
*Cloresmus pulchellus*|XSBN1	0.0020												
*Cloresmus yunnanensis*|XSBN2	0.1230	0.1230											
*Notobitiella bispina*|XSBN3	0.1496	0.1496	0.1447										
*Notobitiella bispina*|XSBN4	0.1513	0.1513	0.1455	0.0013									
*Notobitiella elegans*|XSBN5	0.1246	0.1238	0.1354	0.1264	0.1256								
*Notobitiella elegans*|XSBN6	0.1238	0.1230	0.1354	0.1264	0.1256	0.0020							
*Notobitus meleagris*|YNGN1	0.1349	0.1358	0.1566	0.1466	0.1475	0.1413	0.1413						
*Notobitus meleagris*|LAOS1	0.1300	0.1308	0.1582	0.1526	0.1517	0.1404	0.1404	0.0379					
*Notobitus montanus*|SXQL1	0.1320	0.1320	0.1403	0.1542	0.1542	0.1227	0.1227	0.1338	0.1240				
*Notobitus montanus*|TMS1	0.1320	0.1320	0.1403	0.1517	0.1517	0.1212	0.1212	0.1355	0.1240	0.0033			
*L. membranaceus*|NC_042809	0.1954	0.1954	0.1848	0.1971	0.1980	0.1782	0.1773	0.1937	0.1954	0.1780	0.1771		
*C. tomentosicollis*|KY274846	0.1635	0.1635	0.1689	0.1886	0.1886	0.1558	0.1550	0.1791	0.1791	0.1679	0.1687	0.1817	
*H. longirostris*|EU427337	0.1878	0.1887	0.1853	0.1951	0.1960	0.1767	0.1758	0.1892	0.1805	0.1743	0.1726	0.1730	0.1691

## Data Availability

Data are contained within the article.

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
