# Peer review of "A New Species in Notobitiella Hsiao in China Confirmed by Morphological and Molecular Data (Hemiptera: Heteroptera: Coreidae)"

_insects, 2022, doi:10.3390/insects13050411_

Round 1

Reviewer 1 Report

The manuscript “A new species in Notobitiella Hsiao in China confirmed by morphological and molecular data (Hemiptera: Heteroptera: Coreidae)” (Insects-1656000) by Jiangi and co-authors characterizes and describes a new coreid species. Interestingly, the new species is characterized by its unusual genitalia in comparison to other related species, making it difficult to define if the new species belongs to an already ejected genus or represents a new genus. The species description is based on morophological as well as molecular data (DNA barcoding). In my eyes, the topic of this manuscript is interesting and appropriate for “Insects”. It shows no spectacular results but combines morphological and molecular data as part of a careful species description without much frills. However, there are some parts that have to be modified, changed or added.

It is good to see that the authors use DNA barcodes as part of their species description. Unfortunately, they do not use the Barcode of Life database for their analysis which represents the state-of-the-art workbench of handling DNA barcode data so far (www.boldsystems.org). I strongly recommend that the authors should create a small project at BOLD, using the offered array of tools for analysis. Furthermore, they should think about including the Barcode Index Number (BIN) approach in their study (Ratnasingham and Hebert (2013): PLOS ONE 8: e66213), as it is done in other DNA barcoding studies.

There are various more specific publications which highlight the usefulness of DNA barcoding in terms of true bugs that should be cited, e.g., Park et al. (2011) Barcoding bugs: DNAbased identification of the true bugs (Insecta: Hemiptera: Heteroptera). Public Library of Science ONE 6: e18749, Raupach et al. (2014): Building-up of a DNA barcode library for true bugs (Insecta: Hemiptera: Heteroptera) of Germany reveals taxonomic uncertainties and surprises. Public Library of Science ONE 9 (9): e106940, or Havemann et al. (2018) From water striders to water bugs: The molecular diversity of aquatic Heteroptera (Gerromorpha, Nepomorpha) of Germany based on DNA barcodes. PeerJ 6: e4577.

Be aware that there is no fixed threshold for DNA barcoding to differentiate between species and genus level. I feel it is useful to compare your data with already published data of other true bugs (see above) to validate your findings, beacuse your data does not include all species of the studied tribus.

You use an unrooted tree for your NJ topology. I think the topology should become rooted with an useful outgroup taxon.

See some specific comments made via sticky notes on the PDF file of the manuscript.

Reviewer 2 Report

An article describing a new species of the true bug of the genus Notobitiella. The description of the new species is correct, the photos and figures are very nice and clear. I think the article could be published in 'Insects', but it requires a lot of linguistic changes before it's published. I am enclosing a pdf with suggestions for changes and comments. As I am not native to English, I recommend consulting my changes.

Round 2

Reviewer 1 Report

There are still some few points that have to be corrected and/or added. Take a look on the attached pdf file.

Author Response

Dear Editor and reviewers,

Thank you again for your time to our manuscript. The manuscript has been carefully revised according to your review. Point-by-point responses are included below. We sincerely hope the modification can conform to the request of yours. Thanks again for your reconsideration of our manuscript for publication in your journal.

Point 1: Line 44. Replaced “still be saw as a quite good sequence” to “represents a useful marker”

Response 1: We have replaced “still be saw as a quite good sequence” to “represents a useful marker” in our manuscript.

Point 2: Line 47. Add “even” after “could be”

Response 2: we have added even after “could be” in our manuscript.

Point 3: Line 66. So, the whole genome has been sequenced but not published yet? What about the coverage? How many fragments have been generated?

Response 3: We are sorry to our vague expression. “High-Throughput Sequencing” for samples was thought as “genome-scale” sequencing in some researches [1, 2]. So, we used “whole genome”. After your enlightenment, we noticed this expression could truly confuse people. So, we changed “whole genome” to “mitochondrial genome”. These mitochondrial genomes are not published yet, a total of 2 GB raw data were obtained for each sample. We assembled them by MitoZ, so, the fragments are not covered in the article.

Point 4: Line 71. Did you got complete COI sequences for ALL studied taxa?

Response 4: Yes, we add “for all samples” after “Complete COI gene” in our manuscript.

Point 5: Line 73. complete (?)

Response 5: Yes, we add “complete” before “sequence” in our manuscript.

Point 6: Line 78. .... non-parametric .... [and in contrast to the mentioned correction in the authors reply the reference is still missing (Felsenstein 1994)]

Response 6: we have added “non-parametric” after “1000” and cited (Felsenstein 1985)

Point 7: Line 103. Bootstrap values are still not changed (e.g., 0.725 instead of 0.72 or 72.5%)

Response 7: We have changed our bootstrap value to 72.5% in our manuscript.

Reference

  1. Goubert, C., et al., High-throughput sequencing of transposable element insertions suggests adaptive evolution of the invasive Asian tiger mosquito towards temperate environments. Molecular Ecology, 2017. 26(15): p. 3968-3981.
  2. Zhang, D., et al., Testing the systematic status of Homalictus and Rostrohalictus with weakened cross‐vein groups within Halictini (Hymenoptera: Halictidae) using low‐coverage whole‐genome sequencing. Insect Science, 2022.